# Non-coding RNAs in drug and radiation resistance of bone and soft-tissue sarcoma: a systematic review

**Huan-Huan Chen[1], Tie-Ning Zhang[2], Fang-Yuan Zhang[3]\*, Tao Zhang[2]\***

[1]Department of Oncology, Shengjing Hospital of China Medical University, Shenyang, China; [2]Department of Pediatrics, Shengjing Hospital of China Medical University, Shenyang, China; [3]Department of General Surgery, Shengjing Hospital of China Medical University, Shenyang, China

## Abstract

**Background:** Sarcomas comprise approximately 1% of all human malignancies; treatment resistance is one of the major reasons for the poor prognosis of sarcomas. Accumulating evidence suggests that non-coding RNAs (ncRNAs), including miRNAs, long ncRNAs, and circular RNAs, are important molecules involved in the crosstalk between resistance to chemotherapy, targeted therapy, and radiotherapy via various pathways.

**Methods:** We searched the PubMed (MEDLINE) database for articles regarding sarcoma-associated ncRNAs from inception to August 17, 2022. Studies investigating the roles of host-derived miRNAs, long ncRNAs, and circular RNAs in sarcoma were included. Data relating to the roles of ncRNAs in therapeutic regulation and their applicability as biomarkers for predicting the therapeutic response of sarcomas were extracted. Two independent researchers assessed the quality of the studies using the Würzburg Methodological Quality Score (W-MeQS).

**Results:** Observational studies revealed the ectopic expression of ncRNAs in sarcoma patients who had different responses to antitumor treatments. Experimental studies have confirmed crosstalk between cellular pathways pertinent to chemotherapy, targeted therapy, and radiotherapy resistance. Of the included studies, W-MeQS scores ranged from 3 to 10 (average score = 5.42). Of the 12 articles that investigated ncRNAs as biomarkers, none included a validation cohort. Selective reporting of the sensitivity, specificity, and receiver operating curves was common.

**Conclusions:** Although ncRNAs appear to be good candidates as biomarkers for predicting treatment response and therapeutics for sarcoma, their differential expression across tissues complicates their application. Further research regarding their potential for inhibiting or activating these regulatory molecules to reverse treatment resistance may be useful.

**Funding:** This study's literature retrieval was supported financially by the 345 Talent Project of Shengjing Hospital of China Medical University (M0949 to Tao Zhang).

**\*For correspondence:**
fyzhang@cmu.edu.cn (F-YuanZ);
zhangtaocmu7@126.com (TZ)

**Competing interest:** The authors declare that no competing interests exist.

## Editor's evaluation

Sarcomas comprise approximately 1% of all human malignancies; treatment resistance is one of the major reasons for the poor prognosis of sarcomas. The authors searched the PubMed (MEDLINE) database for articles regarding sarcoma-associated non-coding RNAs from inception to August 17, 2022. Data regarding the roles of ncRNAs in therapeutic regulation and their applicability as biomarkers for predicting the therapeutic response of sarcomas were extracted. The conclusions reached are valuable and solid and should be of interest to the field.

## Introduction

Sarcomas constitute a heterogeneous group of rare tumors representing more than 60 malignancies within a wider range of over 160 different bone and soft tissue neoplasms. Sarcomas include bone sarcoma (osteosarcoma [OS], chondrosarcoma, Ewing's sarcoma [EWS]) and soft-tissue sarcomas (Kaposi's sarcoma, rhabdomyosarcoma [RMS], gastrointestinal stromal tumor [GIST]; *Nacev et al., 2020*). Epidemiologically sarcomas are uncommon, comprising only approximately 1% of human malignancies (*Siegel et al., 2018*), with an estimated annual incidence rate of 2.4 cases per 100,000 population (*Ritter and Bielack, 2010*; *Wibmer et al., 2010*). Treatment strategies include surgical resection, chemotherapy, targeted therapy, immunotherapy, and radiotherapy. Nevertheless, primary or acquired resistance to drugs or radiation eventually leads to treatment failure and poor outcomes in sarcoma patients (*Chen et al., 2016b*; *Wisdom et al., 2020*). The precise mechanisms involved in drug or radiation resistance in sarcoma remain unclear. A better understanding of the mechanisms of sarcoma resistance to drugs or radiation is needed to improve therapeutic efficacy and to prolong the overall survival of patients.

Recently, several lines of evidence have demonstrated that non-coding RNAs (ncRNAs), including miRNAs, long ncRNAs (lncRNAs), and circular RNAs (circRNAs), play vital roles in the resistance to a variety of therapies against bone and soft-tissue sarcomas, including chemotherapy reagents (*Fu et al., 2019*), targeted therapy drugs (*Cao et al., 2016*), immune checkpoint inhibitors (ICIs; *Pang and Hao, 2021*), and radiation (*He et al., 2020*).

miRNAs, which are typically 18–25 nucleotides long, are the most frequently studied short ncRNAs (*Cho, 2010*). Mature miRNAs inhibit target gene expression through mRNA degradation or repression of translation (*Vicente et al., 2016*). By targeting the repressors of specific genes, miRNAs can be used to upregulate the expression of these genes indirectly by alleviating their repression (*Hulshoff et al., 2019*). Increasing evidence has also described a non-canonical role for miRNAs in transcriptional regulation; however, the underlying mechanisms remain largely unknown. ncRNAs comprising over 200 nucleotides are classified as lncRNAs. lncRNAs modulate gene transcription and translation in the cytoplasm through multiple mechanisms (*Chillón and Marcia, 2020*). They can act as recruiters, tethers, and scaffolds for other regulatory factors involved in epigenetic modifications or can regulate gene transcription by acting as decoys, coregulators, or polymerase-II inhibitors. They are also involved in the organization of different components of the transcriptional and splicing machinery and the subnuclear structures. Moreover, lncRNAs can control processes such as mRNA processing, stability, and translation by acting as sponges for miRNAs to block their effects (*Kung et al., 2013*). Finally, circRNAs comprise 1–5 introns or exons and are highly stable molecules that form a covalently closed continuous loop (*Hentze and Preiss, 2013*; *Wilusz and Sharp, 2013*). Similar to lncRNAs, circRNAs also function as miRNA sponges, RNA-binding protein-sequestering factors, and regulators of gene expression by controlling mRNA transcription (*Li et al., 2015*; *Hansen et al., 2013*). Furthermore, circRNAs can control gene transcription by interacting with phosphorylated polymerase-II or by competing with the pre-mRNA splicing machinery (*Zhang et al., 2013*; *Ashwal-Fluss et al., 2014*).

The primary aim of this systematic review was to discuss new paradigms of the roles of regulatory ncRNAs in the molecular mechanisms that underlie the resistance of sarcomas to treatment. The secondary aim of this review was to identify the applicability of ncRNAs as biomarkers for predicting treatment responses, as well as their potential as therapeutic targets.

## Methods

### Search strategy

We searched for relevant articles in PubMed using following Medical Subject Headings: ([non-coding RNA] OR [ncRNA] OR [long non-coding RNA] OR [lncRNA] OR [circular RNA] OR [circRNA] OR [miRNA]) AND ([sarcoma] OR [OS] OR [chondrosarcoma] OR [synovial sarcoma] OR [leiomyosarcoma] OR [liposarcoma] OR [fibrosarcoma] OR [Kaposi's sarcoma] OR [RMS] OR [EWS] OR [angiosarcoma] OR [hemangiosarcoma] OR [GIST] OR [epithelioid sarcoma] OR [alveolar soft part sarcoma] OR [clear cell sarcoma] OR [intimal sarcoma] OR [undifferentiated sarcoma] OR [undifferentiated spindle cell sarcoma] OR [undifferentiated pleomorphic sarcoma] OR [undifferentiated round cell sarcoma] OR [epithelioid inflammatory myofibroblastic sarcoma] OR [myxoinflammatory fibroblastic sarcoma] OR [myofibroblastic sarcoma] OR [ectomesenchymoma] OR [malignant solitary fibrous tumor] OR

[malignant tenosynovial giant cell tumor] OR [epithelioid hemangioendothelioma] OR [malignant glomus tumor] OR [malignant peripheral nerve sheath tumor] OR [malignant granular cell tumor] OR [malignant perineurioma] OR [Neurotrophic tyrosine receptor kinase (NTRK)-rearranged spindle cell neoplasm] OR [desmoplastic small round cell tumor] OR [rhabdoid tumor] OR [desmoid tumor] OR [malignant perivascular epithelioid tumor] OR [malignant ossifying fibromyxoid tumor] OR [myoepithelial carcinoma] OR [malignant mixed tumor] OR [hyalinizing spindle cell tumor] OR [malignant Triton tumor] OR [malignant mesenchymoma]) AND ([resistance] OR [drug resistance] OR [chemoresistance] OR [chemotherapy resistance] OR [radioresistance] OR [radiotherapy resistance] OR [sensitivity]). The reports selected up to August 17, 2022 were included. There were no restrictions with regard to the type of studies or the language used.

### Inclusion and exclusion criteria

Studies fulfilling the following criteria were included: (1) original research studies in which the role of host-derived regulatory ncRNAs (miRNA, lncRNA, or circRNA) in bone or soft-tissue sarcoma was investigated; and (2) studies examining the role of ncRNAs in therapeutic drug or radiation resistance. Investigations on exogenous regulatory RNAs or non-original research articles, such as review articles, conference proceedings, editorials, and book chapters, were excluded.

Titles and abstracts were independently screened for relevance by the two authors of this study (HHC and TZ), while disagreements were resolved through discussions with a third author (FYZ).

### Data extraction

The following data were extracted: first author and year of publication, pathological type of sarcoma, type of study, ncRNA class investigated, therapeutic agents used, methods used to detect the corresponding ncRNA, number of replicates/specimens (for *in vitro* and *in vivo* studies) or patients (for clinical studies), gene or cellular pathways involved, and major conclusions.

### Quality assessment

The quality of the included studies was assessed using the Würzburg Methodological Quality Score (W-MeQS; *Uçeyler et al., 2011*). W-MeQS consists of 12 items that are relevant for assessing the quality of a laboratory method. This tool contains 12 items that assess selection bias, performance bias, attrition bias, detection bias, reporting bias, reagents status, charts status, and measurements status. These factors are commonly involved *in vitro* and *in vivo* studies. The higher the W-MeQS score, the better the quality of the study. For each item fulfilled, one point is given, and the score is the total sum of all points achieved, so the maximum score achievable is 12. If the target of interest was measured by just one method, this score marks the end score. If more than one method was used, a sub-score is calculated for each method as described, and the end score is calculated as the mean of the sub-scores. The quality assessment was independently screened for relevance by two authors (HHC and TZ), and disagreements were resolved through discussions with a third author (FYZ).

## Results

### Summary of included studies and article quality

Our search strategy yielded 930 records. After eliminating duplicates, 927 records remained. Titles and abstracts were screened for content, and 738 underwent a full-text evaluation. The excluded studies were neither original articles nor directly related to sarcoma or lacked evidence of deregulation of the studied ncRNAs in sarcoma (*Supplementary file 3*). The reasons for exclusion and a flowchart of the processing steps are shown in *Figure 1*. A total of 212 original studies investigating miRNAs, lncRNAs, or circRNAs in sarcomas were included. The selected studies were published between 2012 and 2022, with the exception of one published in 2009 and one published in 2010; our selection included *in vitro*, *in vivo*, and human studies. Of these, 178 articles examined the role of ncRNAs in chemotherapy-resistant sarcoma, 14 studied the role of ncRNAs in targeted therapy-resistant sarcoma, one focused on immune checkpoint inhibitor (ICI) resistance, six investigated radiotherapy resistance, and 13 evaluated the value of ncRNAs as biomarkers for predicting treatment response in sarcoma. Of the 212 included studies, 179 examined the role of ncRNAs in OS, 4 focused on chondrosarcoma, 3 on EWS, 2 on synovial sarcoma, and 16 on GIST. Moreover, 135, 54, and 23

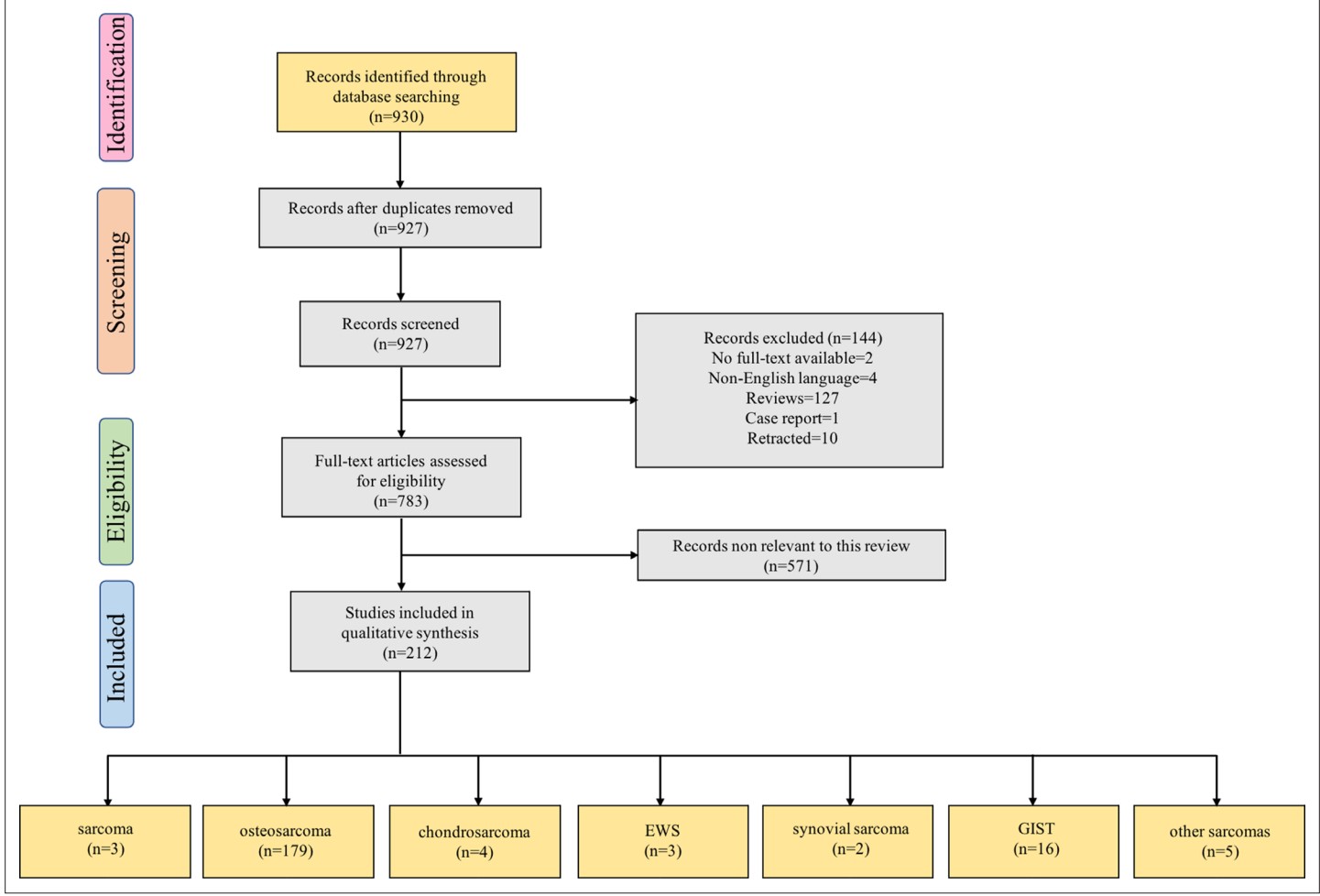

**Figure 1.** Flow diagram for Preferred Reporting Items for Systematic Reviews showing the literature selection process used to identify the studies included in the review. The last group of boxes show the number of studies on different pathological types of sarcomas. Among them, the box 'sarcoma' represents three studies focused on the role of non-coding RNAs in multiple pathological types of sarcoma. Moreover, the box 'other sarcomas' represents five studies focused on rhabdomyosarcoma, uterine leiomyosarcoma, fibrosarcoma, malignant fibrous histiocytoma, and atypical teratoid/rhabdoid tumor. Abbreviations: EWS, Ewing's sarcoma; GIST, gastrointestinal stromal tumor.

studies were focused on miRNAs, lncRNAs, and circRNAs, respectively. *Supplementary file 1* lists all the ncRNAs investigated in different studies. The W-MeQS scores ranged from 3 to 10, with an average score of 5.42 (*Supplementary file 2*), implying that the study quality was compromised for most of the included articles. 95% of the articles had a quality score below 9; only 10 articles had a score of 9 or above. As most of the articles were based on animal or cell experiments, they exhibited obvious distribution bias, selection bias, and reporting bias. A summary of the molecular mechanisms underlying the actions of the ncRNAs associated with therapeutic resistance in sarcoma is shown in *Figure 2* and *Figure 3*.

## ncRNAs participated in chemotherapy drug resistance

### Osteosarcoma

Of the 179 articles that studied the role of ncRNAs in chemotherapy-resistant sarcoma, 166 focused on OS. Of these, 43 investigated the molecular regulatory mechanisms of lncRNAs in chemotherapeutic drug resistance in OS, 101 focused on miRNAs, 21 on circRNAs, and one study reported on both lncRNA MEG3 and hsa_circ_0001258 (*Zhu et al., 2019*). Moreover, 19 overlapping ncRNAs were reported in more than one study, including four lncRNAs (lncRNA SNHG15 [*Sun et al., 2022*; *Zhang et al., 2020a*], lncRNA OIP5-AS1 [*Song et al., 2019*; *Liu and Wang, 2020*; *Sun et al., 2020*; *Kun-Peng et al., 2019*], lncRNA TUG1 [*Zhou et al., 2020*; *Hu et al., 2019*], and

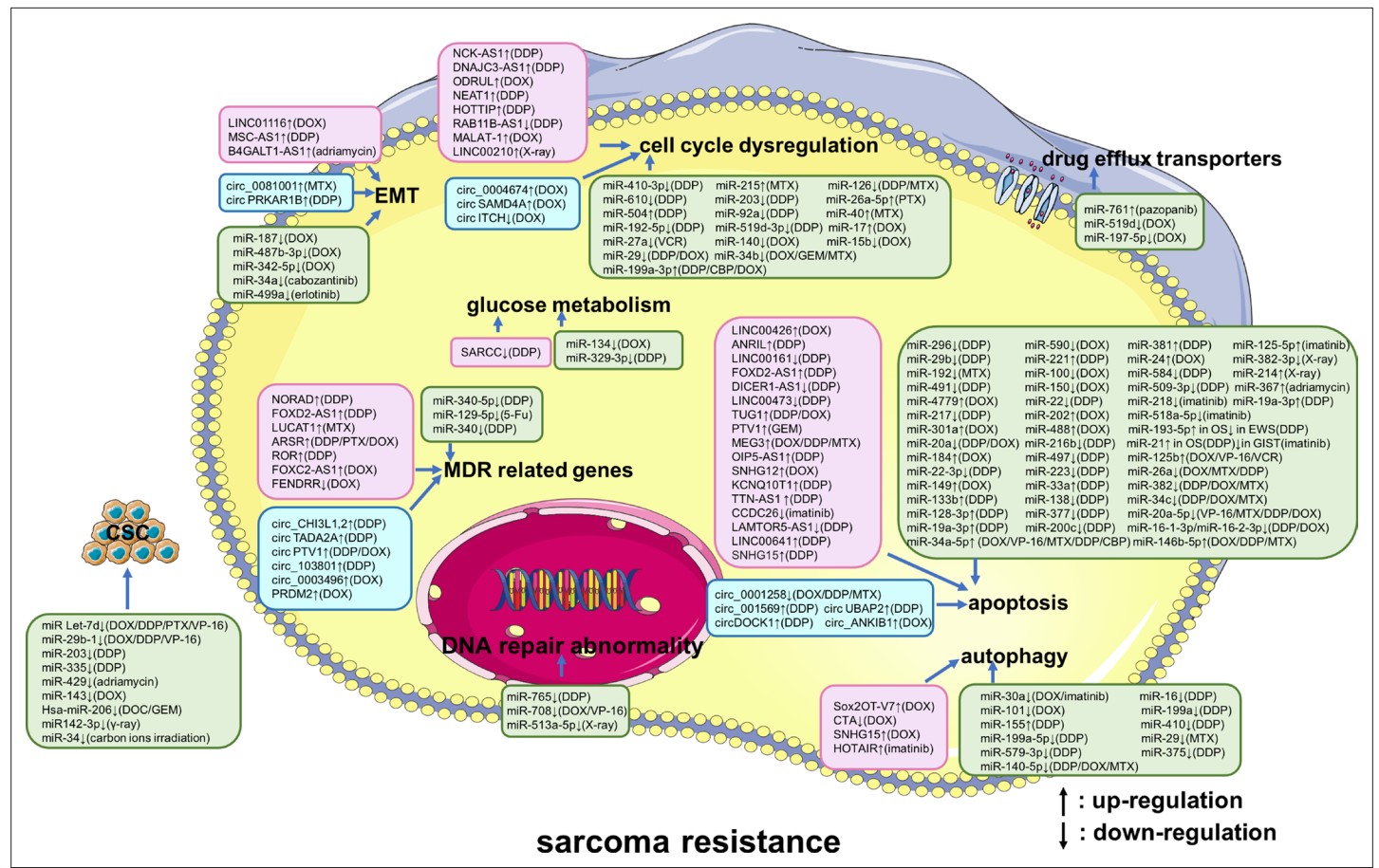

**Figure 2.** A summary diagram of miRNAs, long non-coding RNAs (lncRNAs), and circular RNAs (circRNAs) that participate in drug or radiation resistance in sarcoma. Several miRNAs, lncRNAs, and circRNAs have been found to be involved in sarcoma treatment resistance by influencing apoptosis, DNA repair, the cell cycle, glucose metabolism, autophagy, epithelial-mesenchymal transition, drug efflux, multiple drug resistance, and cancer stem cell behavior, through regulating the expression of potential target genes and related signaling pathways. These phenotypes are disordered in one or more sarcomas of different histological types, including osteosarcoma, chondrosarcoma, Ewing's sarcoma, synovial sarcoma, gastrointestinal stromal tumor, rhabdomyosarcoma, uterine leiomyosarcoma, fibrosarcoma, malignant fibrous histiocytoma, and atypical teratoid/rhabdoid tumors. Specially, these phenotypes are all disordered in osteosarcoma. Abbreviations: 5-Fu, 5-flurouracil; CBP, carboplatin; DDP, cisplatin; DOC, docetaxel; DOX, doxorubicin; GEM, gemcitabine; MTX, methotrexate; PTX, paclitaxel; VCR, vincristine; VP-16, etoposide.

lncRNA ANRIL [*Li and Zhu, 2019b*; *Lee et al., 2021*]), 13 miRNAs (miR-29b [*Luo et al., 2019*; *Li et al., 2020*], miR-21 [*Vanas et al., 2016*; *Ziyan and Yang, 2016*], miR-22 [*Wang et al., 2019c*; *Li et al., 2014*; *Meng et al., 2020a*; *Meng et al., 2020b*; *Zhou et al., 2018b*], miR-199a-3p [*Lei et al., 2018*; *Gao et al., 2015*], miR-34a-5p [*Pu et al., 2016*; *Pu et al., 2017b*; *Pu et al., 2017a*], miR-34a [*Li et al., 2017*; *Novello et al., 2014*; *Chen et al., 2016a*], miR-140-5p [*Wei et al., 2016*; *Meng et al., 2017*], miR-203 [*Chen et al., 2016a*; *Huang et al., 2021*], miR-19a-3p [*Zhang et al., 2019*; *Wang et al., 2022a*], miR-140 [*Song et al., 2009*; *Zhi et al., 2022*], miR-29 [*Osaki et al., 2016*; *Xu et al., 2018b*], miR-221 [*Yu et al., 2019*; *Zhao et al., 2013*], and miR-100 [*Xiao et al., 2017*; *Liu et al., 2016*]), and two circRNAs (circPTV1 [*Li et al., 2021c*; *Kun-Peng et al., 2018a*; *Wang et al., 2022c*] and circRNA_0004674 [*Ma et al., 2021*; *Bai et al., 2021*]) (*Table 1*). The OS studies primarily focused on the molecular mechanisms of ncRNAs in cisplatin resistance (n=79), adriamycin/doxorubicin resistance (n=49), and multidrug resistance (MDR, n=27). The remaining 11 studies evaluated the role of ncRNAs in methotrexate (n=7), etoposide (n=1), paclitaxel (n=1), 5-flurouracil (n=1), and gemcitabine (n=1) resistance in OS. From the 166 studies, we identified 82 ncRNAs that play important roles in enhancing chemoresistance in OS cells and/or animal models. Conversely, 84 ncRNAs had the opposite effect, contributing to chemotherapeutic sensitivity in OS (*Supplementary file 4*).

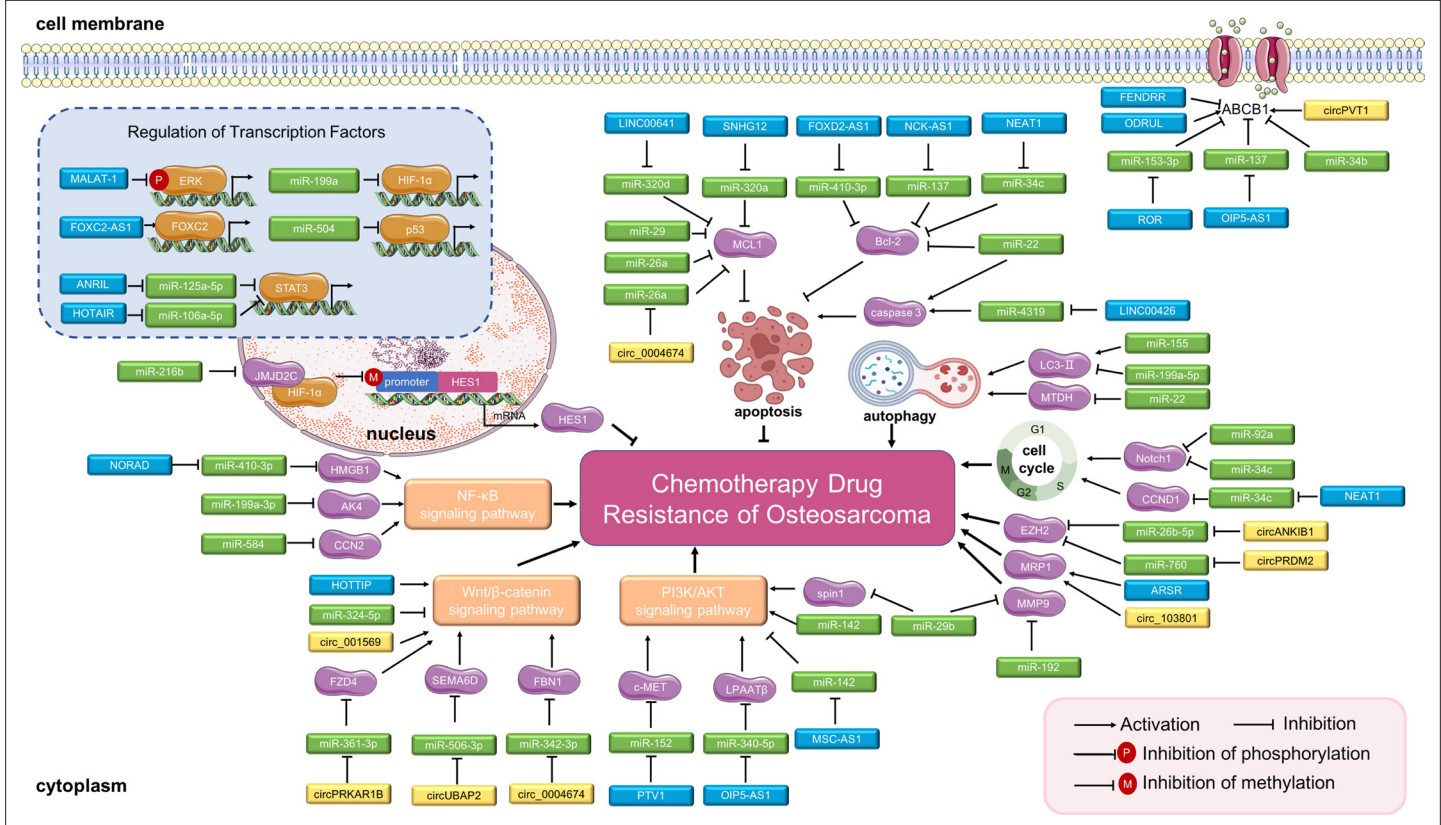

**Figure 3.** Long non-coding RNAs (lncRNAs), circular RNAs (circRNAs), and miRNAs in osteosarcoma chemoresistance. The main molecular mechanisms by which dysregulated ncRNAs (lncRNAs, circRNAs and miRNAs) mediate chemotherapy drug resistance in osteosarcoma are summarized. miRNAs usually bind directly to target genes and regulate their expression and related signaling pathways. LncRNAs and circRNAs can bind directly to target genes or can serve as miRNA sponges to regulate the expression of target genes and related signaling pathways, thereby mediating osteosarcoma chemoresistance. Abbreviations: ABCB1, ATP-binding cassette, subfamily B, member 1; AKT, protein kinase B; CCN2, CTGF, connective tissue growth factor; CCND1, cell cycle-related cyclin D1; ERK, extracellular signal-regulated kinase; EZH2, enhancer of zeste 2 polycomb repressive complex 2; FBN1, fibrillin-1; FOXC2, forkhead box C2; HES1, hairy and enhancer of split-1; HIF-1α, hypoxia-inducible factor-1; LPAATβ, lysophosphatidic acid acyltransferase; MCL1, myeloid cell leukemia 1; MMP-9, matrix metalloproteinase 9; MRP1, multidrug resistance-associated protein-1; MTDH, metadherin; NF-κB, nuclear factor-kappa B; PI3K, phosphoinositide 3-kinase; STAT3, signal transducer and activator of transcription 3.

## EWS and chondrosarcoma

Four studies reported the effects of miRNAs on chemotherapy treatment in EWS. Among these, one study demonstrated that miRNA-193a-5p controls cisplatin chemoresistance in both OS and EWS (*Jacques et al., 2016*). Two studies reported that miRNAs (miR-34a [*Nakatani et al., 2012*] and miR-708 [*Robin et al., 2012*]) increased tumor sensitivity to multiple drugs, including vincristine, doxorubicin, and etoposide, in EWS cells. One study showed that miR-125b overexpression reduced drug sensitivity in EWS cells (*Iida et al., 2013*). In addition, three studies investigated the role of miRNAs in the resistance of chondrosarcoma cells to chemotherapy; the results showed that miR-100 (*Zhu et al., 2014*), miR-23b (*Huang et al., 2017*), and miR-125b (*Tang et al., 2016*) increased cisplatin or doxorubicin sensitivity in chondrosarcoma cells.

## Soft-tissue sarcoma

Among the remaining six studies, five reported that miRNAs (miR-27a [*Bharathy et al., 2018*], miR-22-3p [*Xu et al., 2018a*], miR-34a [*Zhang et al., 2020b*], miR-197-5 p [*Jain et al., 2022*], and miR-206 [*Li et al., 2021a*]) enhanced sensitivity to the chemotherapeutic drugs vincristine, cisplatin, doxorubicin or gemcitabine in RMS, GIST, uterine leiomyosarcoma, and malignant fibrous histiocytoma (MFH), both *in vitro* and *in vivo*. By contrast, one study indicated that miR-17 promotes doxorubicin resistance in synovial sarcoma both *in vitro* and *in vivo* (*Minami et al., 2014*).

**Table 1.** The targets of non-coding RNAs (ncRNAs) that regulate therapeutic resistance in sarcoma.

| Therapeutic strategies | Themes | No. of studies | ncRNA frequently involved | Key genes or pathways involved |
|---|---|---|---|---|
| **Chemotherapy** | | | | |
| | Studies of long ncRNAs (lncRNAs) | 43 | lncRNA SNHG15, lncRNA OIP5-AS1, lncRNA TUG1, and lncRNA ANRIL | NF-$\kappa$B, STAT3, PI3K/AKT, Bax, Bcl-2, caspase3, cleaved caspase3, ABCB1, and MCL1 |
| | Studies on miRNAs | 101 | miR-29b, miR-21, miR-22, miR-199a-3p, miR-34a-5p, miR-34a, miR-140–5 p, miR-203, miR-19a-3p, miR-140, miR-29, miR-221, and miR-100 | MMP-9, KRAS, Bcl-2, PI3K/AKT, NF-$\kappa$B, c-Myc, LC3-$I$, LC3-$II$, HIF-1α, MCL1, North1, Wnt/β-catenin, mTOR, p53, and SOX2 |
| Osteosarcomas (OS) | Studies on circular RNAs (circRNAs) | 21 | circPTV1 and circRNA_0004674 | Wnt/β-catenin, EZH2 |
| Other sarcomas | Studies on ncRNAs | 13 | Various | p53, and AKT |
| **Targeted therapy** | | | | |
| Gastrointestinal stromal tumors (GIST) | Studies on lncRNAs | 4 | lncRNA CCDC26 | |
| | Studies on miRNAs | 6 | miR-125a-5p | Various |
| Other sarcomas | Studies on miRNAs | 4 | Various | Various |
| **Immunotherapy** | | | | |
| Sarcomas | Studies on lncRNAs | 1 | Various | N/A |
| **Radiotherapy** | | | | |
| Sarcomas | Studies on ncRNAs | 6 | Various | Various |
| **Biomarker** | | | | |
| Sarcomas | Studies on ncRNAs | 13 | Various | N/A |

## ncRNAs participated in resistance to targeted therapy drugs

Of the 14 studies that investigated role of ncRNAs in targeted therapy resistance, three focused on the role of miRNAs in OS resistance to targeted therapy drugs. Their *in vitro* and *in vivo* results showed that miR-34a (*Wang et al., 2021*), miR-596 (*Wang et al., 2019a*), and miR-499a (*Wang et al., 2019b*) enhanced the sensitivity of OS to cabozantinib, anlotinib, and erlotinib, respectively. In addition, ten studies illustrated the effects of ncRNAs in GISTs treated with imatinib. Four of the 10 studies demonstrated that the lncRNA CCDC26 (*Cao et al., 2018*; *Yan et al., 2019*) reduced the resistance, whereas HOTAIR (*Zhang et al., 2021a*) and RP11-616M22.7 (*Shao et al., 2021*) enhanced the resistance of GISTs to imatinib *in vitro* and/or *in vivo* by regulating different downstream targets (the c-KIT, ATG2B, IGF-1R, and Hippo signaling pathways). The other six studies reported the roles of different miRNAs in GISTs treated with imatinib: four miRNAs (miR-218 [*Fan et al., 2015*], miR-30a [*Chen et al., 2020a*], miR-518-5p [*Shi et al., 2016*], and miR-21 [*Cao et al., 2016*]) sensitized GIST cells and/or animal models to imatinib, whereas two miRNAs (miR-125-5p [*Huang et al., 2018*; *Akçakaya et al., 2014*] and miR-107 [*Akçakaya et al., 2014*]) enhanced imatinib resistance in GIST cells. Two overlapping ncRNAs, lncRNA CCDC26 (*Cao et al., 2018*; *Yan et al., 2019*) and miR-125-5p (*Huang et al., 2018*; *Akçakaya et al., 2014*), were reported in more than one study. Moreover, the remaining studies showed that miR-761 enhanced pazopanib resistance in synovial sarcoma cells (*Shiozawa et al., 2018*).

## lncRNAs participated in ICI resistance

Only one study met our criteria for ncRNAs in sarcoma resistance to immunotherapy. One bioinformatic analysis reported that the overexpression of the lncRNAs ADAM6, C5orf58, CXCR2P1, FCGR2C, HCP5, HLA-H, NAPSB, NCF1B, and NCF1C reduced the sensitivity of sarcoma to ICIs (*Pang and Hao, 2021*).

## ncRNAs participated in radioresistance

Among the six studies focused on the radiosensitivity of sarcoma, four focused on the sensitivity of OS to X-rays. One study demonstrated that LINC00210 (*He et al., 2020*) reduced the sensitivity of

OS cells to this therapy, and another study showed that miR-214 (*Li et al., 2019a*) induced OS radio-resistance both *in vitro* and *in vivo*. Two studies found that miRNAs (miR-328-3p [*Yang et al., 2018*] and miR-513a-5p [*Dai et al., 2018*]) could directly lead to the radiosensitization of OS cells and animal tumor models. Furthermore, one study reported that rapamycin combined with an miR-34 mimic may overcome the carbon ion irradiation resistance of high-grade chondrosarcoma (*Vares et al., 2020*). The remaining study demonstrated that miR-142–3 p overexpression significantly increased the radio-sensitivity of atypical teratoid/rhabdoid tumor (ATRT) cells (*Lee et al., 2014*).

## ncRNAs as biomarkers for predicting treatment response

Twelve articles evaluated ncRNAs as biomarkers but none of them included an external validation cohort. Six studies identified the role of ncRNAs (lncRNA growth arrest-specific 5 [*Polvani et al., 2022*], lncRNA ENST00000563280 [*Zhu et al., 2015*], miR-21 [*Yuan et al., 2012*], miR-125b [*Luo et al., 2016*], hsa_circ_0008336 [*Han et al., 2020*], hsa_circ_0004664 [*Han et al., 2020*], hsa_circ_0003302 [*Han et al., 2020*], and hsa_circ_0004674 [*Kun-Peng et al., 2018b*]) as biomarkers for predicting chemotherapeutic response in OS. Among the ncRNAs in these studies, the presence of circulating miR-125b was able to distinguish chemotherapy-resistant OS from chemotherapy-sensitive OS with an AUC value of 0.793, and sensitivity and specificity levels of 76.9 and 79.1%, respectively (*Luo et al., 2016*). Moreover, five studies in patients with GISTs focused on differences in the expression of lncRNAs and miRNAs between pre-imatinib/imatinib-sensitive serum or tissues and imatinib-resistant serum or tissues. One of the five reported miR-518e-5p as a potential biomarker for secondary imatinib resistance with an AUC value of 0.9938 and sensitivity and specificity values of 99.8 and 82.1%, respectively (*Kou et al., 2018*). Finally, one study reported that the expression level of the lncRNA HAR1B was higher in pazopanib responders among patients with bone and soft-tissue sarcomas, indicating that HAR1B may serve as a predictive biomarker for response to pazopanib treatment (*Yamada et al., 2021*).

## Target genes, signaling pathways, and functions of ncRNAs

The genes and molecular pathways targeted by ncRNAs that are involved in the regulation of therapeutic resistance in sarcoma are summarized in *Table 1*. The majority of targets identified in the literature search were members of the phosphoinositide 3-kinase/protein kinase B (PI3K/AKT), Wnt/β-catenin, and NF-κB signaling pathways. The most commonly identified targets were ABCB1 (lncRNA LUCAT1 [*Han and Shi, 2018*], lncRNA ODRUL [*Zhang et al., 2016*], lncRNA ROR [*Cheng et al., 2019b*]), lncRNA FENDRR (*Kun Peng et al., 2017*), lncRNA FOXC2-AS1 (*Zhang et al., 2017b*), STAT3 (lncRNA ANRIL [*Li and Zhu, 2019b*]), lncRNA HOTAIR (*Guo et al., 2020*), Bax and Bcl-2 (lncRNA FOXD2-AS1 [*Zhang et al., 2021b*]), lncRNA NCK-AS1 (*Cheng et al., 2019a*), and lncRNA NEAT1 (*Hu et al., 2018*). In addition, ncRNAs targeted MCL1 both directly (lncRNA SNHG12 [*Zhou et al., 2018a*], miR-29 [*Osaki et al., 2016*], and miR-26a [*Li and Ma, 2021b*]) and indirectly (LINC00641 [*Tang et al., 2022b*] via miR-320d), targeted EZH2 both directly (miR-138 [*Zhu et al., 2016*]) and indirectly (circ_ANKIB1 [*Tang et al., 2022a*] via miR-26b-5p), and targeted p53 both directly (miR-34a [*Nakatani et al., 2012*], miR-504 [*Chen et al., 2019*], and miR-125b [*Iida et al., 2013*]) and indirectly (miR-590 [*Long and Lin, 2019*] via ATM). ncRNAs also targeted MMP-9 (miR-29b [*Luo et al., 2019*] and miR-192 [*Bazavar et al., 2020*]), KRAS (miR-192 [*Bazavar et al., 2020*] and miR-217 [*Zhang et al., 2015*]), c-Myc (miR-192 [*Bazavar et al., 2020*] and miR-34a [*Li et al., 2017*]), LC3-Ⅰ and LC3-Ⅱ (miR-101 [*Chang et al., 2014*], miR-155 [*Chen et al., 2014*], and miR-199a-5p [*Li et al., 2016a*]), HIF-1α (miR-199a [*Keremu et al., 2019*] and miR-216b [*Yang et al., 2020*]), North1 (miR-34c [*Xu et al., 2014*] and miR-92a [*Liu et al., 2018*]), SOX2 (miR-429 [*Zhang et al., 2020e*] and miR-29b-1 [*Di Fiore et al., 2014*]), and caspase 3 (LINC00426 [*Wang et al., 2020a*] and the lncRNA NEAT1 [*Hu et al., 2018*]). The majority of commonly expressed ncRNAs were involved in epithelial-to-mesenchymal transition (EMT), cancer stem cells (CSCs), cell cycle dysregulation, glucose metabolism, multi-drug resistance (MDR)-related genes, DNA repair abnormality, apoptosis, autophagy, and drug efflux transporters (*Figure 2* and *Figure 3*).

## Discussion

This systematic review summarized the reports on the molecular mechanisms underlying the involvement of ncRNAs in drug and radiotherapy resistance in bone and soft-tissue sarcomas published over the past decade. Our aim, to identify common factors in the dysregulation of ncRNAs that are associated with therapeutic resistance in sarcoma, may provide clues about the the mechanisms underlying this resistance. However, only a few overlapping ncRNAs were reported in two or more studies focusing on chemotherapy resistance in OS and targeted therapy resistance in GIST. Moreover, we found that the downstream targets of ncRNAs were predominantly found in the PI3K/AKT, Wnt/β-catenin, and NF-κB pathways. We also described the differential expression of ncRNAs between chemoresistant and chemosensitive sarcoma tissues and/or cells, which may support the potential of ncRNAs as biomarkers for predicting the sarcoma effect. Here, we further discuss some of the key findings from the studies included in this review and delineate some major limitations and potential prospects.

### ncRNAs in chemoresistance of OS

High-dose methotrexate with leukovorin-rescue, ifosfamide, doxorubicin, and cisplatin are considered the most active agents for OS treatment (*Cortes et al., 1974*; *Smrke et al., 2021*), but drug resistance is still the main cause of disease progression and recurrence. Resistance to chemotherapy in sarcoma can be linked to perturbations of the mechanisms that underlie signal transduction, cell death (apoptosis and autophagy), MDR-related gene expression, and transcriptional factor regulation (*Lilienthal and Herold, 2020*). There is evidence supporting the role of perturbed signal transduction pathways, such as the PI3K/AKT, Wnt/β-catenin, and NF-κB pathways, in the development of chemotherapy resistance in OS. It has been shown that components of the PI3K/AKT signaling pathway are frequently altered in human cancers and that these changes may contribute decisively to the resistant phenotype (*Fresno Vara et al., 2004*). ncRNAs affect the sensitivity of OS cells by targeting PI3K/AKT signaling directly or indirectly. c-MET, a receptor for hepatocyte growth factor, has been reported to promote tumorigenicity in a variety of cancers (*Wu et al., 2017*). For instance, lncRNAAPTV1 promoted gemcitabine resistance in OS cells by activating PI3K/AKT signaling via c-MET (*Sun et al., 2019*). MiR-221 also promoted cisplatin resistance in OS cells by directly activating PI3K/AKT (*Zhao et al., 2013*). Spindlin 1 (spin1), a new member of the SPIN/SSTY family, has been shown to promote tumorigenesis in human cancers (*Fang et al., 2018*). By contrast, miR-29b increased doxorubicin sensitivity in OS through inhibition of PI3K/AKT signaling by regulating spin1 expression (*Li et al., 2020*). Moreover, aberrant activation of Wnt/β-catenin signaling is tightly linked with therapy response in various cancers (*Zhang and Wang, 2020c*). Studies included in our review also demonstrated that ncRNAs regulate resistance to chemotherapy drugs by regulating Wnt/β-catenin signaling. For example, lncRNA HOTTIP (*Li et al., 2016b*), circ_001569 (*Zhang et al., 2018*), and circPRKAR1B (*Feng et al., 2021*) promoted drug resistance by activating Wnt/β-catenin signaling. Among these ncRNAs, the lncRNA HOTTIP (*Li et al., 2016b*) and circ_001569 (*Zhang et al., 2018*) targeted Wnt/β-catenin directly. Frizzled class receptor 4 (FZD4), a class Frizzled G-protein-coupled receptor (GPCR), is also a WNT receptor (*El-Sehemy et al., 2020*). CircPRKAR1 activated Wnt/β-catenin by sponging miR-361–3 p, and thus upregulated the expression of FZD4 (*Feng et al., 2021*). By contrast, miR-342–5 p sensitized OS cells to doxorubicin by inhibiting the expression of both Wnt7b and β-catenin (*Liu et al., 2019*). In addition, emerging studies indicate that dysregulation of the NF-κB pathway causes cancers (*Yu et al., 2020*) and enhances drug resistance in tumor cells (*Mirzaei et al., 2022*). The ncRNAs can potentially function as upstream mediators and modulate NF-κB in OS. At the molecular level, *Xie et al., 2020* indicated that the lncRNA NORAD (Non-coding RNA-activated by DNA damage) promoted the cisplatin resistance of OS by sponging miR-410–3 p and thus activating NF-κB. By contrast, miR-410–3 p promoted the drug sensitivity in OS cells by downregulating high-mobility group box-1 (HMGB1) and, subsequently, inhibiting NF-κB activity (*Wang et al., 2020b*).

In addition to the abnormal activation of key signal transduction pathways, alterations in cell death signaling may also contribute to chemotherapy resistance in OS. ncRNAs have been found to participate in OS chemoresistance by regulating apoptosis-related proteins, such as Bcl-2, Bax, caspase 3, and MCL1. It has been reported that overexpression of the anti-apoptotic protein Bcl-2 in malignant cells fortifies their drug-resistance capacity (*Hafezi and Rahmani, 2021*). Conversely, the upregulation of the pro-apoptotic proteins Bax and caspase-3, and of the activated form cleaved caspase-3, plays the opposite role in human cancers (*Hafezi and Rahmani, 2021*; *Fulda, 2015*).

For instance, the lncRNA NCK-AS1 was found to be upregulated in cisplatin-resistance OS cell lines. At the molecular level, the lncRNA NCK-AS1 enhanced drug resistance in OS by upregulating Bcl-2 and downregulating Bax and cleaved caspase-3 through the sponging of miR-137 (*Cheng et al., 2019a*). *Hu et al., 2018* reported that the lncRNA NEAT1 also promoted cisplatin-resistance in both OS cells and a xenograft model by upregulating Bcl-2 and downregulating Bax via miR-34c. Myeloid cell leukemia-1 (MCL1), an antiapoptotic member of the BCL2 family, contributes to cell survival and resistance to diverse chemotherapeutic agents in human cancers (*Sancho et al., 2022*). The study conducted by *Zhou et al., 2018a* showed that the lncRNA SNHG12 enhanced doxorubicin-resistance in OS cells by positively regulating the expression of MCL1 by sponging miR-320a. Moreover, circ_0004674 facilitated OS progression and doxorubicin resistance by upregulating MCL1 via miR-142-5p (*Ma et al., 2021*). By contrast, miR-26a reversed MDR in OS cells, and when xenografted in nude mice, it directly inhibited the expression of MCL1 (*Li and Ma, 2021b*). In addition, the regulation of autophagy-related proteins, including HMGB1, LC3-Ⅰ, and LC3-Ⅱ, by ncRNAs also played important roles in OS chemoresistance. The chromatin-binding nuclear protein HMGB1 plays a role in facilitating autophagy following the administration of cytotoxic agents (*Tang et al., 2010*). It has been reported that overexpression of HMGB1 in OS cell lines allowed them to resist autophagy when treated with doxorubicin, cisplatin, and methotrexate (*Huang et al., 2012*). In addition, the overexpression of miR-22 hindered doxorubicin and cisplatin resistance in OS by inhibiting HMGB1-promoted autophagy *in vitro* (*Li et al., 2014*). LC3 and LC3 homologs enable autophagosomes that have the ability to bind autophagic substrates and/or proteins that mediate cargo selectivity (*Galluzzi and Green, 2019*), and it has been proved that the overexpression of LC3-Ⅱ promoted autophagy and caused drug resistance in cancers (*Wang et al., 2022b*). *Chen et al., 2014* found that overexpression of miR-155 significantly enhanced the conversion of LC3-I to LC3-II, promoted autophagy, and enhanced chemoresistance in OS. Conversely, miR-199–5 p reduced cisplatin resistance in OS cells by downregulating LC3-Ⅱ and reducing the ratio of LC3-Ⅱ to LC3-Ⅰ (*Li et al., 2016a*).

Furthermore, abnormal expression of MDR-related proteins plays an important role in OS chemoresistance. The studies included in our review described the regulatory effect of ncRNAs on drug-resistance-related proteins, especially ATP-binding cassette subfamily B member 1 (ABCB-1), in OS. ABCB-1, a member of the ABC family of efflux transporters, is a classical MDR-related protein (*Franke et al., 2010*). The lncRNA FOXC2-AS1 may promote doxorubicin resistance in OS by facilitating ABCB1 expression via increasing the expression of the transcription factor FOXC2 (*Zhang et al., 2017b*). Overexpression of circPVT1 contributed to the doxorubicin and cisplatin resistance of OS cells by positively regulating ABCB-1 (*Kun-Peng et al., 2018a*). Conversely, miR-34b reversed drug resistance in OS by directly lowering the expression of ABCB-1 (*Zhou et al., 2016*). In addition, transcription factors such asp53 and HIF-1α could be present downstream of ncRNAs and have been found to be involved in OS chemoresistance. Wild-type p53 is central for maintaining genomic stability and preventing oncogenesis, whereas mutant p53 is tightly associated with late-stage malignance and drug resistance in cancers (*Zhou et al., 2019*). Wild-type p53-induced phosphatase 1 (WIP1), an oncogene that is overexpressed in diverse cancers, has been regarded as a critical inhibitor of the ataxia telangiectasia mutated (ATM)/radiation resistance gene 3 related (ATR)-p53DNA damage signaling pathway (*Lu et al., 2008*). In OS cells, miR-590 inhibited doxorubicin resistance by negatively regulating WIP1, and subsequently reduced the expression of both ATM and p53 (*Long and Lin, 2019*). It has been proved that the HIF family of hypoxia-inducible transcription factors is widely upregulated in human cancers. HIF-1α has been associated with chemotherapy failure in various cancers (*Rohwer and Cramer, 2011*). Histone demethylase jumonji C domain-containing 2 C (JMJD2C) has been shown to serve as a co-activator for HIF-1α in cancer progression (*Luo et al., 2012*). *Yang et al., 2020* found that miR-216b enhanced cisplatin sensitivity in OS cells by downregulating JMJD2C and HIF-1α, inhibiting the expression of the hairy and enhancer of split-1 (HES1) gene.

Collectively, ncRNAs play a key role in regulating OS chemoresistance, and therapeutic strategies that are based on small-molecule activator or inhibitor ncRNAs have the potential to rescue therapeutic resistance in patients with OS. Nevertheless, only a limited number of studies have illustrated the molecular mechanisms underlying the roles of ncRNAs in regulating OS chemoresistance; this aspect needs further research.

## miRNAs in chemoresistance in EWS, chondrosarcoma, and soft-tissue sarcomas

EWS is an aggressive sarcoma of the bone and soft tissue that occurs at any age and has a 5-year overall survival rate of 65–75% for patients with localized disease and of <30% for those with metastases, except for those with isolated pulmonary metastasis for whom the 5-year survival rate is approximately 50% (*Gaspar et al., 2015*). Recent evidence highlighted that miRNAs were involved in various tumor processes related to chemoresistance in EWS, especially in regulating p53. For example, overexpression of miR-125b is associated with the downregulation of the pro-apoptotic molecules p53 and Bak, resulting in enhanced drug resistance in EWS (*Iida et al., 2013*). By contrast, a study performed by *Nakatani et al., 2012* demonstrated that overexpression of miR-34a in wild-type p53 EWS cells decreased malignancy and increased tumor sensitivity in response to doxorubicin and vincristine.

Chondrosarcoma is the second most common primary malignant bone sarcoma, and usually exhibits resistance to chemotherapy (*Zając et al., 2021*). miRNAs have been associated with reduced drug resistance in chondrosarcoma. For instance, the study conducted by *Huang et al., 2017* showed that miR-23b increased cisplatin sensitivity in chondrosarcoma by inhibiting the Src-Akt pathway. Moreover, *Tang et al., 2016* demonstrated that miR-125b acted as a tumor suppressor in chondrosarcoma cells by increasing doxorubicin sensitization by directly targeting the oncogene ErbB2, leading to the inhibition of glucose metabolism.

Soft-tissue sarcomas are rare tumors that account for 1% of all adult malignancies, with over 100 different histologic subtypes occurring predominantly in the trunk, extremities, and retroperitoneum (*Gamboa et al., 2020*). The studies included in this review found that miRNAs participated in regulating chemoresistance in several soft-tissue sarcomas, such as RMS, fibrosarcoma and MFH. For instance, the PAX3:FOXO1 fusion oncogene mediated tolerance to chemotherapy in RMS (*Singh et al., 2022*). *Bharathy et al., 2018* reported that overexpression of miR-27a led to PAX3:FOXO1 mRNA destabilization and chemotherapy sensitization in RMS both *in vitro* and *in vivo*. In fibrosarcoma, miR-197–5 p sensitizes HT1080 cells to doxorubicin by suppressing the expression of MDR genes, ABCC1, and major vault protein (MVP; *Jain et al., 2022*). Furthermore, miR-206 showed low levels of expression in docetaxel-resistant MFH cells. Mechanistically, miR-206 significantly inhibited MFH proliferative activity by regulating the properties of CSCs (*Li et al., 2021a*).

In summary, several biological mechanisms underlying bone and soft-tissue sarcomas involve ncRNAs, indicating that ncRNAs may be targets for therapies intended to overcome or prevent chemoresistance in sarcomas. For example, the inhibition of oncogenic ncRNAs and the activation of tumor-suppressive ncRNAs are promising therapeutic strategies for sarcoma treatment. Furthermore, a combination of targeted ncRNA therapy with conventional chemotherapy may effectively reverse sarcoma drug resistance and significantly improve the effects of chemotherapeutics, subsequently improving prognosis. Nevertheless, the studies focused on the regulatory role of ncRNAs in soft-tissue sarcoma are very limited. Thus, the implementation of therapies targeting specific ncRNAs to overcome drug resistance in sarcomas remains a challenge, and most ncRNAs have not been characterized for potential clinical applications. Therefore, further investigation and clinical trials are required to develop novel ncRNA-related therapeutic strategies to overcome drug resistance in sarcoma.

## ncRNAs in resistance to targeted therapies in GIST and OS

GISTs are the most common subtype of soft-tissue sarcoma (*Ducimetière et al., 2011*). The targeted therapy drug imatinib is the gold standard therapy for GIST. This treatment has been found to prolong patient survival effectively, especially in the high-risk GIST group, as shown by a randomized trial study that included follow-up of 9.1 years (*Casali, 2021*). Therefore, imatinib resistance in GISTs is an important factor in disease progression and relapse. Recent studies have identified a vital role for ncRNAs in imatinib resistance in GIST. For example, ncRNAs may serve upstream of autophagy-related proteins (autophagy-related protein 2 homolog B [ATG2B] and Beclin1) in GIST and thus regulate drug sensitivity. At the molecular level, the lncRNA HOTAIR activated autophagy and promoted the imatinib resistance of GIST cells by increasing the expression of ATG2B via miR-130a (*Zhang et al., 2021a*). Inversely, miR-30a inactivated autophagy and sensitized GIST cells to imatinib by downregulating Beclin1 (*Chen et al., 2020a*). Moreover, it has been shown that members of the non-receptor protein tyrosine phosphatase (PTPN) family are differentially expressed in digestive tract cancers and are closely associated with improved disease prognosis (*Chen et al., 2020b*). With regard

to GIST, *Akçakaya et al., 2014* found that miR-125a-5p enhanced imatinib resistance by suppressing the expression of PTPN18.

In addition to the use of imatinib in GIST, targeted therapeutic drugs, the antiangiogenic agents anlotinib (*Wang et al., 2019a*), cabozantinib (*Wang et al., 2021*), and pazopanib (*Shiozawa et al., 2018*), and the epidermal growth factor receptor (EGFR) inhibitor erlotinib (*Wang et al., 2019b*) have also been used as rescue drugs after chemotherapy failure in OS. Several studies have investigated the mechanisms of resistance to targeted therapy involving ncRNAs in OS. For example, the Notch pathway is major regulator in human malignancies and also mediates drug resistance in cancer cells (*Zhang et al., 2017a*). *Wang et al., 2021* reported that miR-34a overexpression promoted the sensitivity of OS to cabozantinib by suppressing the Notch pathway. Moreover, SH3KBP1-binding protein 1 (SHKBP1) is an upstream molecule of EGFR that prevents EGFR degradation (*Liu et al., 2022*). The study conducted by *Wang et al., 2019b* found that the TGFβ–miR-499a–SHKBP1 network orchestrates the EMT-associated kinase switch, which induces resistance to erlotinib in CD166[+] OS CSC-like cells.

Overall, targeting of lncRNAs and miRNAs may be a promising strategy to improve the efficacy of sarcoma-targeted therapies. There have, however, been increasing concerns regarding the therapeutic potential of targeting a single ncRNA and other current targeting strategies. First, despite the numerous studies that have attempted to reveal the mechanisms and effects of ncRNAs, we have gained only superficial knowledge in the field, and the effects of circRNAs in sarcoma-targeted therapy remain largely unclear. Furthermore, given the large number of ncRNAs and their upregulation or downregulation in sarcoma, it is crucial to determine the most clinically relevant ncRNAs that have the greatest impact on disease outcome. Therefore, further studies are needed to investigate the mechanisms of ncRNA action in sarcoma-targeted drug resistance in order to improve patient survival.

## lncRNAs in resistance of sarcomas to immunotherapy

Immunotherapy, which has been used for melanoma, is a new therapy paradigm that holds great promise for sarcoma treatment (*Falcone et al., 2020*; *Rutkowski et al., 2020*). Although the anti-CTLA-4 drug ipilimumab was well tolerated by patients with synovial sarcoma, no obvious efficacy was observed in this patient group (*Maki et al., 2013*). However, in a separate study using the anti-PD-1 antibody SARC028 to treat patients with bone sarcomas, a partial response was observed (NCT02301039; *Tawbi et al., 2017*). The primary resistance to immunotherapy may be caused by an immunosuppressive environment in which there is no pre-existing antitumor response (*Saleh and Elkord, 2020*). The overexpression of ICI molecules may create an immunosuppressive environment, leading to immunotherapy resistance in sarcoma. A previous study had found that the elevated expression of several lncRNAs was correlated with reduced immune cell infiltration and reduced sensitivity to ICIs in sarcomas (*Pang and Hao, 2021*). Despite disappointing results from preliminary immunotherapy trials for sarcomas, the combination of ICIs with cytotoxic chemotherapies or targeted therapies may significantly improve the prognosis for sarcoma patients.

Overall, immunotherapy is a promising strategy that requires specific adjustments for use in patients with sarcomas. More in-depth studies are needed to clarify the association between abnormal ncRNA expression and clinical efficacy of ICIs in sarcomas; for example, further investigations are needed to understand the changes in expression of ncRNAs that occur in the tumor tissue or serum of sarcoma patients treated with ICIs. Both *in vitro* and *in vivo* experiments are needed to investigate the molecular mechanisms underlying the influence of ncRNAs on ICI treatment in sarcomas. ncRNAs hold the potential to be used in predicting drug sensitivity and in improving treatment efficacy through the detection or perturbation of their expression, respectively, in patients with sarcoma who are undergoing immunotherapy.

## ncRNAs in resistance of OS, chondrosarcoma, and ATRT to radiotherapy

Pre- and postoperative radiotherapy is widely used in sarcomas (*Haas, 2014*). Radiotherapy leads to the activation of an interconnected series of processes in the tumor microenvironment, including inflammation, cycling hypoxia, immunomodulation, revascularization, extracellular matrix remodeling coordinated by cancer-associated fibroblasts, and fibrosis. These changes affect the radiosensitivity

of cancer cells (*Barker et al., 2015*). Moreover, many mechanisms for CSC radioresistance have been proposed, including drug efflux through ABC transporters, overactivation of the DNA damage response, apoptosis evasion, activation of the prosurvival pathway, cell cycle promotion, and/or cell metabolic alterations (*Garcia-Mayea et al., 2020*). In addition, the role of ncRNAs in radiotherapy resistance has been investigated in hepatocellular carcinoma, breast cancer, lung cancer, and gastric cancer (*Zhang et al., 2020d*). Several studies have illustrated the molecular mechanisms through which ncRNAs regulate radiosensitivity in OS. For example, *He et al., 2020* demonstrated that knockdown of the lncRNA LINC00210 enhances the radiosensitivity of OS cells by acting as a miR-342–3 p sponge to positively regulate expression of the oncogene GDNF receptor alpha 1 (GFRA1). By contrast, overexpression of miR-328–3 p enhances the radiosensitivity of OS cells by directly targeting histone H2AX (*Yang et al., 2018*). The effect of miR-34 and miR-142–3 p in radioresistance in chondrosarcoma (*Vares et al., 2020*) and ATRT (*Lee et al., 2014*) have been identified by regulating the transcription factors FOXO3 and SOX2, respectively. Specifically, overexpression of miR-34a enables chondrosarcoma cells to overcome resistance to carbon-ion irradiation by upregulating FOXO3, which leads to KLF4 repression (*Vares et al., 2020*). Moreover, miR-142-3p was shown to reduce γ radiation resistance in pediatric brain ATRT by inhibiting expression of sex-determining region Y box 2 (SOX2) (*Lee et al., 2014*).

In summary, targeting ncRNAs to overcome radiotherapy resistance in sarcomas is gaining interest; however, the number of studies cited in our review is still small. Further research is needed to explore the influence of additional ncRNAs on sarcoma radiotherapy. To select the most clinically effective target ncRNAs for reversing radiotherapy resistance, more in-depth *in vitro* and/or *in vivo* experimental studies as well as multi-center clinical studies with large samples, are urgently needed. In the future, interfering with the expression of ncRNAs may become an important strategy for improving the sensitivity of radiotherapy and the prognosis of patients with sarcoma.

## ncRNAs as biomarkers for treatment monitoring

Sarcomas lack specific tumor markers. Although elevated lactate dehydrogenase levels, or more frequently alkaline phosphatase levels, in the serum have been found in some patients with OS, these serum indicators lack specificity and sensitivity. Therefore, the diagnosis and prognostic evaluation of diseases depend on imaging examinations and invasive biopsies (*Ritter and Bielack, 2010*). These methods are inconvenient for both primary screening of the disease and monitoring of short-term treatment outcomes. Hence, non-invasive biomarkers that have high sensitivity and specificity are required. Recently, exosomes, including tumor-associated proteins, enzymes, growth factors, bioactive lipids, miRNAs, and DNA sequences, have been considered as potential biomarkers for sarcoma diagnosis and prognosis evaluation, and as possible targets for sarcoma therapy (*Min et al., 2016*). The differential expression of ncRNAs between drug-resistant and drug-sensitive sarcoma tissues or cells demonstrates the applicability of ncRNAs as biomarkers for prediction of treatment effect. In particular, a previous study reported that miR-518e-5p serves as a biomarker for imatinib resistance, showing showed high sensitivity (99.8%) and specificity (82.1%) with an AUC value of 0.9938 in GISTs (*Kou et al., 2018*). A separate study reported an AUC value of 0.793 for miR-125b as a biomarker for predicting the chemosensitivity of OS, with a sensitivity of 76.9% and a specificity of 79.1%.

Overall, ncRNAs may have potential to serve as biomarkers for sarcoma drug resistance and to predict therapeutic responses in patients with sarcoma. However, because ncRNA detection can vary immensely depending on the method used, in-depth studies should focus on optimizing ncRNA detection methodology. Importantly, the results of the studies cited in this review may be biased, as most were single-center trials with small sample sizes. Therefore, further studies are needed to accelerate the clinical application of ncRNAs, and the inclusion of multi-center research studies will be particularly important in reducing the errors caused by differences between individual centers. In addition, the studies included in this review are mainly focused on lncRNAs and miRNAs, highlighting the lack of understanding of the role of circRNAs as biomarkers for the prediction of treatment effect in sarcoma and the need for further investigations. We expect future analysis of tumor-specific ncRNA biomarkers to offer not only improved diagnosis but also a convenient and sensitive method for monitoring the outcomes of treatments in sarcoma patients.

## Conclusions

Despite the variability in the results of the studies examined, this systematic review supports the notion that ncRNAs have potential to be used as ideal biomarker candidates for treatment monitoring and future therapeutic targets in sarcomas. With further research, ncRNAs may become powerful compounds for sensitizing therapy-resistant sarcomas to standard treatments.

## Acknowledgements

We would like to acknowledge the PubMed (MEDLINE) databases that supplied the literature retrieval library.

## Additional information

### Funding

| Funder | Grant reference number | Author |
|---|---|---|
| 345 Talent of Shengjing Hospital of China Medical University | | Tao Zhang |

The funders had no role in study design, data collection and interpretation, or the decision to submit the work for publication.

### Author contributions

Huan-Huan Chen, Data curation, Writing - original draft, Writing - review and editing; Tie-Ning Zhang, Data curation, Software; Fang-Yuan Zhang, Conceptualization, Data curation, Software; Tao Zhang, Conceptualization, Data curation, Methodology, Writing - original draft

### Author ORCIDs

Fang-Yuan Zhang ![ORCID] http://orcid.org/0000-0002-1551-6048
Tao Zhang ![ORCID] http://orcid.org/0000-0001-5341-8249

### Decision letter and Author response

Decision letter https://doi.org/10.7554/eLife.79655.sa1
Author response https://doi.org/10.7554/eLife.79655.sa2

## Additional files

### Supplementary files

• Supplementary file 1. Summary of non-coding RNAs in sarcoma therapeutic resistance.

• Supplementary file 2. Quality assessment of the included studies according to Würzburg Methodological Quality Score (W-MeQS).

• Supplementary file 3. References of studies excluded in the systematic review (N=715).

• Supplementary file 4. Description of the 212 original studies of non-coding RNAs in sarcoma.

• MDAR checklist

### Data availability

All data generated or analysed during this study are included in the manuscript and supporting file. The data has also been deposited to Dryad.

The following dataset was generated:

| Author(s) | Year | Dataset title | Dataset URL | Database and Identifier |
|---|---|---|---|---|
| Zhang T, Chen H, Zhang T, Zhang F | 2022 | 212 orginal articles | https://dx.doi.org/10.5061/dryad.kd51c5b8t | Dryad Digital Repository, 10.5061/dryad.kd51c5b8t |

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
