## [Editor Report]

Sarcomas comprise approximately 1% of all human malignancies; treatment resistance is one of the major reasons for the poor prognosis of sarcomas. The authors searched the PubMed (MEDLINE) database for articles regarding sarcoma-associated non-coding RNAs from inception to August 17, 2022. Data regarding the roles of ncRNAs in therapeutic regulation and their applicability as biomarkers for predicting the therapeutic response of sarcomas were extracted. The conclusions reached are valuable and solid and should be of interest to the field.

---

## [Decision Letter]

**Decision letter after peer review:**

Thank you for submitting your article "Non-coding RNAs in drug and radiation resistance of bone and soft tissue sarcoma: A systematic review" for consideration by *eLife*. Your article has been reviewed by 2 peer reviewers, and the evaluation has been overseen by a Reviewing Editor and Mone Zaidi as the Senior Editor. The following individuals involved in the review of your submission have agreed to reveal their identity: Danh Dinh Truong (Reviewer #1); John Prensner (Reviewer #2).

Essential revisions:

1) Referees' suggestions should be addressed, particularly regarding the scope to achieve general appeal.

2) The authors should devote some effort to connecting the literature compiled to a broader scope, as to how it relates or does not relate to other types of cancer.

*Reviewer #1 (Recommendations for the authors):*

This manuscript provides a comprehensive overview of the Non-coding RNA landscape in soft tissue and bone sarcoma. The authors detail how non-coding RNAs affect different treatment options. Their work used a search algorithm to find relevant articles studying non-coding RNAs in sarcomas. This resulted in an expansive table that included details on the studies, such as histopathology, non-coding RNA investigated, methods, and summary. In addition, the authors provide their interpretation and overview divided by type of treatments. The strengths of this manuscript include expanded search algorithm, an overview of each research article included, and interpretation of the collected articles. The work will likely impact the field as a consensus review article for how non-coding RNA could affect sarcoma treatment. The tables are especially useful for those interested. Overall, the work represents a significant undertaking to collect and summarize the current studies on non-coding RNA and sarcoma.

Overall, the work is comprehensive. However, there are some concerns prior to acceptance.

1. There are a few articles listed here that already cover some of the materials. What is the contribution of the reviewed manuscript compared to these others?

Min, Li, et al. "Potentials of long noncoding RNAs (LncRNAs) in sarcoma: from biomarkers to therapeutic targets." International journal of molecular sciences 18.4 (2017): 731.

Li, Xiaoyang, et al. "Noncoding RNA in drug-resistant sarcoma." Oncotarget 8.40 (2017): 69086.

2. While sarcomas are very diverse with more than 50 subtypes, much of the focus is on osteosarcoma. There does not even seem to be a definitive non-coding RNA even for osteosarcoma. The authors mention various non-coding RNAs but the review would highly benefit from a better synthesis to where the authors can synthesize various papers and come up with a conclusion for a few definitive non-coding RNAs that affect osteosarcoma treatment. It is also not useful to describe sarcoma as a whole since their biologies are quite vast.

3. I suggest to the reviewers that they re-evaluate the articles and re-organize based on pathways and non-coding RNAs. In its current form, it is difficult to parse out important non-coding RNAs. Potential readers would first be interested in a sarcoma subtype and pathway, but in its current shape, it is organized by treatment modality, which makes it difficult to parse out the sarcoma subtype and pathway.

4. Figure 2 summarizes all of the non-coding RNA is misleading since not all sarcomas are the same histology. Are all sarcomas affected similarly by the non-coding RNA? Are the authors suggesting that these pathways are dysregulated in all sarcomas?

5. The study also seems to limit the keywords based on the inclusion of sarcoma. However, not all sarcomas have the word 'sarcoma' in the name, for example, desmoplastic round cell tumor, epithelioid hemangioendothelioma, desmoid tumor, Malignant peripheral nerve sheath tumor.

6. While Osteosarcoma seems to be the most prevalent finding, the authors would benefit from delineating the various osteosarcoma type. There are various types of osteosarcoma, such as osteoblastic, fibroblastic, chondroblastic, etc., but it is not clear which are studied in the paper or if it is defined at all.

7. It is not clear how the SYRCLE was being used. Is this just a measure the author calculates and leaves to readers to determine the quality of an article? Or was it used for any filtering purposes in Figure 1?

*Reviewer #2 (Recommendations for the authors):*

The systematic review by Chen and colleagues investigates the available literature for non-coding RNAs in radiation resistance in sarcomas. In this article, the authors apply Cochrane Review/SYRCLE methodology to assess the literature currently available for sarcomas. Although the authors dutifully compile a large amount of cited literature, this review ultimately fails to capture my interest for several fundamental reasons, which are detailed below. Overall, I find the review shallow in its critical thinking, questionable in its methodology, and unclear as to the significance or magnitude of the problem being addressed.

Comments

1. At the highest level, it is not clear why a review dedicated to non-coding RNAs in drug and radiation resistance of bone and soft tissue sarcomas is the most critical topic for the authors to focus on. While it is true that non-coding RNAs are generally an area of significant interest for cancer biology, this review is based on a very narrow slice of cancer. Why not consider cancers generally? If the primary interest is non-coding RNAs in cancer, then restricting to sarcomas seems unjustified. If the primary interest is in clinical biomarkers for sarcomas, then restricting to non-coding RNAs also seems unjustified.

2. The methodology applied by the authors is not designed for this type of analysis of the scientific literature. Therefore, the authors use an inappropriate method to attempt to derive conclusions. Cochrane reviews are specifically designed to assess clinical trials in a systematic way. SYRCLE (Hooijmans, BMC Medical Research Methodology, 2014) is specifically designed to assess animal studies. The vast majority of non-coding RNA research is benchtop molecular biology. While the authors state that they adapt SYRCLE to fit their needs, it is not clear to me that these adaptions are sufficient to make it an appropriate methodology. Therefore, the authors' approach has irreconcilable flaws.

To detail one aspect of this, one of the fundamental goals of systematic reviews of clinical trials is to compare the effect size of the clinical phenotype across studies, and it does this by also taking into account the number of patients in the trial. Therefore, a trial with 10,000 patients gets more analytical weight than a trial with 100 patients. There simply is no real comparable application to the basic scientific literature. Essentially, the closest comparator would be giving extra emphasis to the highest-quality papers compared to anecdotal or partial investigations. But the authors do exactly the opposite by essentially considering all papers to be equally valid, despite the widely variable amount and quality of data in them. This gives the false impression that this review is unbiased but a more accurate description would be that it dilutes the important papers by giving the questionable papers equal attention.

3. Related to #2, this review lacks a compelling synthesis of the research field. There are many pieces of literature cited (many which probably should not be cited). But there is no grand vision of the major technological, clinical, and research advances and challenges that lay ahead. For example, Figure 2 shows a graphic of various non-coding RNAs participating in various aspects of cancer cell biology, but it is not clear whether these all represent robust observations versus single reports in the literature of variable data quality.

4. I would recommend broadening the scope of the article to review non-coding RNAs as diagnostics in adult cancers.

5. I would recommend not using the SYRCLE methodology.

6. I would recommend determining whether the primary goal of this review is to evaluate the biology of non-coding RNAs or a clinical biomarker phenotype for non-coding RNAs, as these are different and will require different considerations.

---

## [Author Response]

Reviewer #1 (Recommendations for the authors):This manuscript provides a comprehensive overview of the Non-coding RNA landscape in soft tissue and bone sarcoma. The authors detail how non-coding RNAs affect different treatment options. Their work used a search algorithm to find relevant articles studying non-coding RNAs in sarcomas. This resulted in an expansive table that included details on the studies, such as histopathology, non-coding RNA investigated, methods, and summary. In addition, the authors provide their interpretation and overview divided by type of treatments. The strengths of this manuscript include expanded search algorithm, an overview of each research article included, and interpretation of the collected articles. The work will likely impact the field as a consensus review article for how non-coding RNA could affect sarcoma treatment. The tables are especially useful for those interested. Overall, the work represents a significant undertaking to collect and summarize the current studies on non-coding RNA and sarcoma.Overall, the work is comprehensive. However, there are some concerns prior to acceptance.1. There are a few articles listed here that already cover some of the materials. What is the contribution of the reviewed manuscript compared to these others?Min, Li, et al. "Potentials of long noncoding RNAs (LncRNAs) in sarcoma: from biomarkers to therapeutic targets." International journal of molecular sciences 18.4 (2017): 731.Li, Xiaoyang, et al. "Noncoding RNA in drug-resistant sarcoma." Oncotarget 8.40 (2017): 69086.

In recent years, studies about noncoding RNA in various diseases has been increasing quickly. We searched PubMed database for articles on non-coding RNAs relevant to sarcoma from inception to August 17, 2022, collected 212 studies in our research, compared with the two articles listed above, the number of included articles increased significantly. Second, our study focused on all the noncoding RNAs, not only lncRNAs (first article listed above), but also miRNAs and circRNAs. Third, Li, Xiaoyang, et al., studied the ncRNAs participated in chemotherapy drug resistance, our article studies ncRNAs participated in chemotherapy drug resistance, targeted therapy drug resistance, immune checkpoint inhibito resistance and radio resistance, as well as the ncRNAs as biomarkers for predicting treatment response. In conclusion, compared with past studies, our study covered a wider range of research subjects, and the data is more comprehensive.

2. While sarcomas are very diverse with more than 50 subtypes, much of the focus is on osteosarcoma. There does not even seem to be a definitive non-coding RNA even for osteosarcoma. The authors mention various non-coding RNAs but the review would highly benefit from a better synthesis to where the authors can synthesize various papers and come up with a conclusion for a few definitive non-coding RNAs that affect osteosarcoma treatment. It is also not useful to describe sarcoma as a whole since their biologies are quite vast.

We re-evaluated the articles and summarized several overlapping non-coding RNAs and genes that affect osteosarcoma treatment, respectively, in Table 1. However, only a few non-coding RNAs are overlapped. We considered that it might not be suitable to organize this paper based on non-coding RNAs among osteosarcoma or other sarcomas. Thus, we re-organized this paper based on important genes or pathways of different histopathological types, respectively. We wonder if this existing form are appropriate for potential readers?

3. I suggest to the reviewers that they re-evaluate the articles and re-organize based on pathways and non-coding RNAs. In its current form, it is difficult to parse out important non-coding RNAs. Potential readers would first be interested in a sarcoma subtype and pathway, but in its current shape, it is organized by treatment modality, which makes it difficult to parse out the sarcoma subtype and pathway.

Related to 2, we summarized important non-coding RNAs and pathways in Table 1. We also re-organized this paper based on sarcoma subtypes and pathways. If there are any shortcomings in the current shape, we are very willing to continue to improve it.

4. Figure 2 summarizes all of the non-coding RNA is misleading since not all sarcomas are the same histology. Are all sarcomas affected similarly by the non-coding RNA? Are the authors suggesting that these pathways are dysregulated in all sarcomas?

To reduce misunderstanding, we explained the phenotypes that affected by non-coding RNAs during the processes of treatment resistance of different histopathological types of sarcomas in the figure legend of Figure 2. Moreover, since much of the focus is on osteosarcoma, we made Figure 3 to further describe the main molecular mechanisms by which dysregulated non-coding RNAs mediated osteosarcoma chemotherapy drug resistance.

5. The study also seems to limit the keywords based on the inclusion of sarcoma. However, not all sarcomas have the word 'sarcoma' in the name, for example, desmoplastic round cell tumor, epithelioid hemangioendothelioma, desmoid tumor, Malignant peripheral nerve sheath tumor.

We have changed the search strategy included as many sarcoma types as possible in accordance with the WHO classification of soft tissue neoplasms, 2013 and 2020 versions. And searched for relevant articles in PubMed using Medical Subject Heading (MeSH), the strategy was listed in “methods”.

6. While Osteosarcoma seems to be the most prevalent finding, the authors would benefit from delineating the various osteosarcoma type. There are various types of osteosarcoma, such as osteoblastic, fibroblastic, chondroblastic, etc., but it is not clear which are studied in the paper or if it is defined at all.

We have added a new column in supplementary Table 1 and delineated the various osteosarcoma subtype in all the 212 included articles, but only a few mentioned the classification of osteosarcoma, we listed them in supplementary Table 1, because the number was too small, we can't do a deeper analysis.

7. It is not clear how the SYRCLE was being used. Is this just a measure the author calculates and leaves to readers to determine the quality of an article? Or was it used for any filtering purposes in Figure 1?

The SYRCLE was a measurement that the authors used to assess the quality of included articles, not used for filtering in Figure 1. The readers can know the quality of the included literature and the reliability of the conclusion through the score table, and we also focused on the literature with high quality score in the discussion.

Actually, there was no standardized rating scale to assess the quality of laboratory methods in scientific papers. Refer to the reviewer's comments and some previous studies, we chose Würzburg Methodological Quality Score (W-MeQS) as the new scoring tool, which consists of twelve items that are relevant for high quality of a laboratory method. This tool contains twelve items assessing selection bias, performance bias, attrition bias, detection bias, reporting bias, reagents status, charts status and measurements status. These factors are common in vitro and vivo studies. If the target of interest was measured by just one method this score marks the end score. If more than one method was used, a subscore is calculated for each method as described and the end-score is calculated as mean of the subscores. In conclusion, the scoring system of W-MeQS is more complete than SYRCLE, would reflect the quality of the article better in a certain degree.

Reviewer #2 (Recommendations for the authors):1. I would recommend broadening the scope of the article to review non-coding RNAs as diagnostics in adult cancers.

This is a very good suggestion for systematic understanding of the diagnostic value of non-coding RNA in adult cancer. But this is a huge project, studies about noncoding RNA in cancer has been increase quickly in recent years. We tried to search the relevant literature in PubMed and found that there were more than 42,000 articles.

Our team focuses on the clinical and the mechanisms of drug resistance in sarcoma, so we selectively study the non-coding RNAs in drug and radiation resistance of bone and soft tissue sarcoma, the primary aim was to discuss new paradigms of the roles of regulatory ncRNAs in the molecular mechanisms underlying sarcoma treatment resistance, secondary aim was to identify the applicability of ncRNAs as biomarkers for predicting treatment response, and their potential as therapeutic targets.

In order to refine our research, we have changed the search strategy included as many sarcoma types as possible in accordance with the WHO classification of soft tissue neoplasms, 2013 and 2020 versions. And searched for relevant articles in PubMed using Medical Subject Heading (MeSH), collected 212 studies in our revised manuscript. Compared with some past studies which focused on noncoding RNA in drug-resistant sarcoma, our study focused on all the type of noncoding RNAs participated in chemotherapy drug resistance, targeted therapy drug resistance, immune checkpoint inhibitor resistance and radio-resistance in sarcoma, as well as the noncoding RNAs as biomarkers for predicting treatment response. The study covered a wider range of research subjects, and the data is more comprehensive, would give readers some inspiration.

2. I would recommend not using the SYRCLE methodology.

We reviewed all the articles that have been included in our research (including those newly included in our study), most were animal or cell line experiments, there were no RCT study or observational study, or non-randomized controlled experimental study have been included in our systemic review.

Actually, there was no standardized rating scale to assess the quality of laboratory methods in scientific papers. Through some previous reports, we chose Würzburg Methodological Quality Score (W-MeQS) as the new scoring tool, which consists of twelve items that are relevant for high quality of a laboratory method. This tool contains twelve items assessing selection bias, performance bias, attrition bias, detection bias, reporting bias, reagents status, charts status and measurements status. These factors are common in vitro and vivo studies, If the target of interest was measured by just one method this score marks the end score. If more than one method was used, a subscore is calculated for each method as described and the end-score is calculated as mean of the subscores. In conclusion, the scoring system of W-MeQS is more complete than SYRCLE, would reflect the quality of the article better in a certain degree.

3. I would recommend determining whether the primary goal of this review is to evaluate the biology of non-coding RNAs or a clinical biomarker phenotype for non-coding RNAs, as these are different and will require different considerations.

After expanding the search, we total included 212 articles in this review. Among them, 200 articles demonstrated that the roles of non-coding RNAs in the molecular mechanisms underlying sarcoma treatment resistance. 12 articles identify the applicability of ncRNAs as biomarkers for predicting treatment response. Hence, the primary aim of this review is to evaluate the biology of non-coding RNAs in treatment resistance (Figure 2 and Figure 3) and the secondary aim of this review is to evaluate non-coding RNAs as clinical biomarkers for monitoring treatment response (We also pointed these two aims in the last paragraph of “introduction”).